# Carbon Ion Irradiation Activates Anti-Cancer Immunity

**DOI:** 10.3390/ijms25052830

**Published:** 2024-02-29

**Authors:** Makoto Sudo, Hiroko Tsutsui, Jiro Fujimoto

**Affiliations:** 1Department of Gastroenterological Surgery, Hyogo Medical University, Nishinomiya 663-8501, Japan; ma-sudou@hyo-med.ac.jp (M.S.); sfujimo@hyo-med.ac.jp (J.F.); 2Osaka Heavy Ion Therapy Center, Osaka 540-0008, Japan

**Keywords:** carbon ion radiotherapy (CIRT), anti-cancer immunity, cancer immunosurveillance, tumor microenvironment (TME), regulatory T cells (Tregs), myeloid-derived suppressor cells (MDSCs), tumor-associated macrophages (TAMs)

## Abstract

Carbon ion beams have the unique property of higher linear energy transfer, which causes clustered damage of DNA, impacting the cell repair system. This sometimes triggers apoptosis and the release in the cytoplasm of damaged DNA, leading to type I interferon (IFN) secretion via the activation of the cyclic GMP–AMP synthase-stimulator of interferon genes pathway. Dendritic cells phagocytize dead cancer cells and damaged DNA derived from injured cancer cells, which together activate dendritic cells to present cancer-derived antigens to antigen-specific T cells in the lymph nodes. Thus, carbon ion radiation therapy (CIRT) activates anti-cancer immunity. However, cancer is protected by the tumor microenvironment (TME), which consists of pro-cancerous immune cells, such as regulatory T cells, myeloid-derived suppressor cells, and tumor-associated macrophages. The TME is too robust to be destroyed by the CIRT-mediated anti-cancer immunity. Various modalities targeting regulatory T cells, myeloid-derived suppressor cells, and tumor-associated macrophages have been developed. Preclinical studies have shown that CIRT-mediated anti-cancer immunity exerts its effects in the presence of these modalities. In this review article, we provide an overview of CIRT-mediated anti-cancer immunity, with a particular focus on recently identified means of targeting the TME.

## 1. Introduction

Many types of solid cancers develop deep inside the body and frequently face hypoxia due to rich stromal cells and/or poor vasculatures for their mass size. A conventional photon beam attenuates rapidly after entering the body. As adverse effects on the entering site limit the dose of photon irradiation, only a small dose of the beam reaches the cancer site [1,2]. Furthermore, even if it reaches the cancerous mass, the photon beam cannot generate reactive oxygen species (ROS), which are critical tumoricidal molecules, under hypoxic conditions [1,2]. This may indicate that some solid cancers are resistant to photon radiation therapy. Ion-beam radiation generated from heavy atoms has been developed to overcome the properties of photon radiation [1,2]. Based on the physical properties, carbon ion beams can reach deep sites, and the higher linear energy transfer, which is a unique property of carbon ion beams, induces clustered DNA breaks that activate the cell repair system. Moreover, the spread-out Bragg peak, which is a unique property commonly shared by particle ions including carbon ions, allows healthy tissues beneath the cancer mass to evade high levels of the ion particle beams. Furthermore, ROS can be generated independently of the presence of oxygen. Clinical studies have demonstrated successful treatment with carbon ion radiotherapy (CIRT) in patients with various types of cancer. The properties of heavy ion beams have recently been reported in detail [1,2,3,4]. 

Murine cancer-bearing athymic nude mice lacking thymus-derived T-cells are resistant to CIRT, whereas euthymic wild-type mice are sensitive [5]. It was reported that the elimination of actions of regulatory T cells (Treg) that negatively regulate T cell functions, enhanced the anti-cancer ability of CIRT [6,7]. These results allowed us to speculate on the importance of T cells in the anti-cancer activities of CIRT. In this article, we provide an overview of the mechanisms underlying the CIRT induction of anti-cancer actions via the activation of host anti-cancer immunity, both innate and adaptive, with a particular focus on recent findings. We also describe the possible theoretical application of CIRT in collaboration with other novel anti-cancer strategies to strengthen its cancer-cure potential.

## 2. Cancer Cell-Killing Modules of Carbon Ion Beam

### 2.1. Direct Cancer Cell-Killing Action

Carbon ion irradiation causes heavy DNA lesions consisting of clustered double-strand (ds) and single-strand DNA breaks in the nuclear DNA of cancer cells [1,4,8,9]. DNA lesions induce cell cycle arrest. This is frequently followed by misrepair or the absence of repair, which eventually results in the apoptotic death of cancer cells and occasionally cancer stem cells [10]. Upon the carbon ion exposure of cancer cells, DNA breaks occur in the mitochondrial DNA [11]. Because mitochondrial DNA encodes genes essential for mitochondrial function, DNA breaks in mitochondrial DNA cause mitochondrial dysfunction, which eventually results in impaired cellular metabolism, such as poor ATP generation. Thus, heavy-ion irradiation directly kills cancer cells via DNA breaks in nuclear DNA and mitochondrial DNA. 

### 2.2. Mitochondria-Mediated Apoptosis

The carbon ion induction of mitochondrial dysfunction also induces cytochrome c release, which then activates the apoptotic caspase cascade; cytochrome c activates caspase 9, and active caspase-9 then activates caspase 7/8, which subsequently activates caspase-3. Active caspase-3 results in apoptotic cancer cell death. The precise molecular mechanisms underlying carbon ion irradiation-induced apoptotic cancer cell death and other types of cell death have been reported in recent review articles [11]. 

## 3. Cancer Immunosurveillance

The concept of cancer immunosurveillance was proposed more than a hundred years ago, but the concept was just like “a fairy tale” without evidence [12]. It took several decades or more to understand the complicated immune system, understand the intimate crosstalk between the immune system and other systems, such as the nervous system, and generate machinery and tools to address hypotheses based on scientific settings [13,14,15,16]. These conditions allowed us to conduct cancer immunosurveillance.

Throughout the body, cellular mutations occur frequently and stochastically, and mutated cells, including cancer cells, are deleted by various intracellular and intercellular management systems. The immune system was clearly demonstrated to be necessarily required for the elimination of mutated cells [13,14,15]. Early studies reported that mice deficient in adaptive immune cells spontaneously develop cancer: mice deficient in gene recombination-activating gene (Rag)-2, which is key to generating T cells and B cells, namely, *Rag2*^−/−^ mice; severe immune-deficient (SCID) mice lacking both T cells and B cells; and mice deficient in T cell receptor α and β-chains, named *αβ T*^−/−^ mice, are vulnerable to cancer development [17,18,19]. This was also the case for mice deficient in the cytocidal modules of natural killer and cytotoxic T cells, such as perforin, and mice harboring impaired cytocidal modules, including interferon (IFN)-γ signaling [20,21,22,23]. Thus, adaptive immune cells play an essential role in the prompt elimination of spontaneously developing cancer cells. 

Natural killer (NK) cells are believed to contribute to cancer immunosurveillance because mice deficient in NKp46, a major activation receptor, spontaneously develop cancer [24]. However, the molecular mechanisms underlying the NK cell-mediated death of cancer cells remain unknown. Very recently, it was reported that human and murine NK cells recognize externalized calreticulin in cancer cells under endoplasmic reticulum (ER) stress via NKp46, a major activating NK receptor, followed by the killing of cancer cells. Human NK cells deficient in *NCR1*, which encodes NKp46, lack this activity [25]. Thus, NK cells, or NKp46-expressing cells, play an important role in cancer immunosurveillance.

Once they are equipped with immune-regulatory arms, such as the expression of programmed cell death ligand-1 and CD47, and the potential to generate robust immunosuppressive and physical barriers, called the tumor microenvironment (TME) and consisting of immunoregulatory cells and perhaps a dense stroma, cancer cells overcome host immunosurveillance and develop into a cancer mass [26] (Figure 1). We previously assumed that immunosurveillance-evading cancer cells never return to immunosurveillance-susceptible cells. However, we have acquired immune checkpoint inhibitors (ICIs) [27,28] and are still on the way to further reverse this process. This may contribute to the complete elimination of immunosurveillance-evading cancer.

We now know that impairments or dysfunctions in systemic physiological conditions, such as dysbiosis, might also participate in cancer evasion during immunosurveillance [29,30,31,32]. 

### 3.1. Immunoregulatory Cells in Immunosurveillance-Evading Cancer

The immunoregulatory dominance of TME is a prominent feature of immunosurveillance-evading cancer and is involved in cancer growth and metastasis. The cellular constituents of the TME differ according to cancer type. Regulatory T cells (Tregs), myeloid-derived suppressor cells (MDSCs), and tumor-associated macrophages (TAMs) are commonly found in various cancers. Recent review articles have clearly shown that cancer-associated gene mutations trigger the corresponding TME generation [33,34,35,36,37]. 

#### 3.1.1. Tregs

Tregs express forkhead box P3 (Foxp3), a master regulator of Tregs. Tregs exert immunosuppressive effects through several mechanisms. First, cytotoxic T-lymphocyte associated protein 4 (CTLA4) on Tregs prevents conventional T-cell activation. Upon stimulation with antigens, conventional T-cells recognize antigen-derived peptides in association with major histocompatibility complex (MHC) expressed on antigen-presenting cells (APCs) via T cell receptors and bind to co-stimulators, namely, CD80 or CD86, expressed on APCs via CD28. Treg-derived CTLA4 protects against the activation of conventional T cells by interrupting CD28-mediated signaling. Second, Tregs consume high levels of interleukin (IL)-2. Conventional T cells also require IL-2. Thus, the remaining IL-2 is insufficient for the complete activation of conventional T cells. Third, the interaction between CTLA4 and CD80 or CD86 induces programmed cell death 1 ligand 1 (PD-L1) release from APCs. PD-L1 suppresses conventional T-cells via programmed cell death receptor 1 (PD-1) interactions. Fourth, Tregs produce several types of inhibitory cytokines such as IL-10 and tumor growth factor β. Currently, CTLA4 is a target for cancer treatment [33,35]. 

Tregs express Foxp3, CTLA4, IL-2Rα (CD25), glucocorticoid-induced tumor necrosis factor (TNF) R-related protein, T-cell immunoreceptor with Ig and immunoreceptor tyrosine-based inhibitory motif domains, carbon catabolite repression 4, and PD-1 [33]. To dampen Tregs in cancer, inhibitors of these molecules, known as ICIs, have been developed and used in cancer immunotherapy.

Tregs are necessary for preventing autoimmune diseases [33,35]. Various types of autoimmune diseases develop severely if humans and mice are deficient in molecules essential for Tregs, such as Foxp3, CTLA4, IL-2Rα (CD25), glucocorticoid-induced TNFR-related protein, T-cell immunoreceptor with Ig and ITIM domains, carbon catabolite repression 4, and PD-1 [33]. Therefore, it is plausible that the depletion or malfunction of Tregs during cancer therapy produces adverse effects, as exemplified by autoimmune diseases. Modalities targeting cancer-specific Treg cells without affecting autoimmune regulatory Treg functions will solve this issue and will be the next generation of regimens for cancer immunotherapy. 

Very recently, to identify master regulators selectively expressed in cancer-infiltrating Tregs, Obradovic et al. performed a comparative study of master regulators between tumor-infiltrating Tregs and peripheral Tregs, and identified 17 master regulators as functional determinants of tumor-infiltrating Treg transcripts. They performed CRISPR-Cas9 screening in vivo and identified *Trps1* as a master regulator of ectopic cancer growth using CRISPR knockout screening. Mice bearing *Trps1* sgRNA, but not those bearing scrambled sgRNA, were resistant to ectopic cancer. In contrast, mice bearing TRPS1-sgRNA evade autoimmunopathology in peripheral tissues, including the skin, colon, small intestine, liver, and kidney [38]. This study may be regarded as an initiator that opens a window for selectively targeting Tregs in the TME.

#### 3.1.2. MDSCs

MDSCs are major components of the TME. MDSCs develop from the myeloid lineage and are composed of two types of cells: neutrophil-related polymorphonuclear MDSCs derived from granulocytic precursors and monocyte-related MDSCs derived from monocytic precursors. These two cellular populations have different morphologies, activating stimuli, phenotypic surface markers, transcriptional regulation, developing factors, and tools for suppressing T cells; however, they are equally preventing cancer from anti-cancer immunity, including T, B, and NK cells. Recent review articles summarized the characteristics of MDSCs [36,37,39,40]. 

It is well established that MDSCs are involved in cancer metastasis that pivotally determines the fate of patients with cancer. MDSCs participate in the preparation of premetastatic niches after leaving the cancer and entering the circulation [41]. Once MDSCs reach premetastatic niches using their chemokine receptors, they prime niches by generating ROS, producing inflammatory cytokines, and remodeling the matrix [42,43]. In particular, neutrophil-related polymorphonuclear MDSCs contribute to niche preparation by releasing neutrophil extracellular traps. Neutrophil extracellular traps were originally reported as bactericidal weapons that activate neutrophils. However, the generation of neutrophil extracellular traps is also involved in cancer metastasis [44,45,46,47]. Upon inoculation with cancer cells, mice deficient in the ability to generate neutrophil extracellular traps or mice treated with inhibitors of neutrophil extracellular traps exhibit less spontaneous metastasis than control mice [44,45]. Circulating cancer cells released from cancer masses are the seeds of metastasis [39]. MDSCs secrete circulating cancer cells to maintain their potential to develop into metastatic niches [48]. Thus, MDSCs contribute to the protection of cancer cells and their metastasis from immunosurveillance.

Various modalities for MDSCs have been developed based on their characteristics [36]. Herein, we briefly introduce recent developments in cancer therapy targeting MDSCs. A decade ago, it was reported that MDSCs, both neutrophil-related polymorphonuclear MDSCs and monocyte-related MDSCs, derived from tumors of cancer-bearing mice, have a shorter life span than macrophages and neutrophils [49]. MDSCs are vulnerable to spontaneous apoptosis. MDSCs robustly express an apoptosis-inducing receptor designated death receptor 5, alternatively named TNF-related apoptosis-inducing receptor 2 [49,50,51]. Cancer-derived MDSCs sampled from patients with cancer showed increased death receptor 5 expression in response to ER stress. A clinical study revealed that the treatment of patients with advanced solid tumors with an agonistic monoclonal antibody (mAb) for death receptor 5 diminished MDSCs in the circulation and that patients with a reduction of MDSCs to the control levels took a much longer time to progress than those resistant to the treatment [50]. These studies suggest the potential of death receptor 5 agonists in cancer immunotherapy by reducing MDSCs in the TME. Furthermore, a mouse cancer model revealed that death receptor 5 agonists and ICIs synergistically protect against cancer [51]. Thus, death receptor 5 agonists appear to be potent cancer immunotherapeutic agents, particularly in combination with ICIs.

Recently, IL4 blockade in the bone marrow prevented murine lung cancer growth by inhibiting MDSC development [52]. This was also true for patients with non-small-cell lung cancer. Macrophages in the lung are developed from two types of cells: tissue-resident macrophages that arise during embryonic development and renewal in the lung, and monocyte-derived macrophages that are derived from hematopoietic stem cells and differentiate via granulocyte-monocyte progenitors in the bone marrow. LaMarche et al. performed gene set enrichment analysis between monocyte-derived macrophages (mo-mac) infiltrating the non-small-cell lung cancer of patients and tissue-resident macrophages of healthy lungs, and found that IL-4-signaling was the most highly enriched among mo-mac-specific genes. They verified similar results in melanoma metastasis-bearing mice and in a genetic mouse model of lung cancer, *Kras^G12D^Tp53^−/−^* lung adenocarcinoma. Treatment with a neutralizing anti-IL-4 mAb inhibited cancer growth in the cancer-bearing mice and the genetic mouse model. Mice selectively deficient in the IL-4 receptor I in tissue-resident macrophages showed cancer growth comparable to that in wild-type mice, suggesting the indispensable role of tissue-resident macrophages IL-4 signaling in cancer expansion. In contrast, mice deficient in IL-4R in the granulocyte-monocyte progenitors exhibited poorer cancer growth than wild-type mice and had much smaller numbers of mo-mac in cancer, suggesting that IL-4 signaling is a prerequisite for the development of MDCSs. Nonetheless, these mice harbored a substantial number of neutrophils and mo-mac. Based on these results, the researchers concluded that IL-4 signaling is required for the development of MDSCs, but not neutrophils or mo-mac. Finally, they revealed that the anti-IL-4R mAb synergizes with the anti-PD1/programmed cell death ligand 1 mAb to protect against non-small-cell lung carcinoma. Thus, IL-4 blockade may become a standard regimen for anti-cancer therapy to prevent the development of MDCSs.

#### 3.1.3. TAMs

Macrophages are divided into two populations based on their origin: embryo-derived tissue-resident macrophages, such as Kupffer cells in the liver and alveolar macrophages in the lungs; and hematopoietic stem cells in the bone marrow via monocyte development [53,54,55]. Macrophages exhibit a diverse range of functions. Macrophages are classified into two types based on their phenotype: M1 and M2. M1 macrophages participate in inflammatory responses by producing proinflammatory cytokines and engulfing, phagocytizing, and killing target cells. In contrast, M2 macrophages contribute to tissue repair and cancer progression. Macrophages are characterized by plasticity, which is highly tuned by their milieu. Thus, the macrophage phenotypes are labile. TAMs have similar fundamental characteristics. This may imply that the extrinsic transfer of long-lasting macrophages with fixed proinflammatory features is more suitable for TAM-targeting therapy than rewiring endogenous TAMs into the M1 type. Recently, second-generation M1-polarized chimeric antigen receptor (CAR) macrophages were generated and reported to have superior anti-cancer functions in liver cancer-bearing mice [56]. Second-generation CAR construction is composed of an extracellular anti-cancer domain, a cytoplasmic TIR domain that is essential for signaling and shared by IL-1R, IL-18R, and Toll-like receptors, and the CD3ζ signal transduction domain. The TIR domain drives M1-like macrophage polarization and induces efferocytosis in dead cancer cells. After inoculation with liver cancer cells, mice treated with macrophages containing truncated CAR lacking both the TIR and CD3ζ domains showed robust cancer development and died within 70 days after inoculation. In contrast, mice treated with full-armed CAR macrophages evaded cancers and survived until day 100. Thus, second-generation CAR macrophages might be a breakthrough in cancer immunotherapy.

### 3.2. Anti-Phagocytic Molecule CD47 on Cancer Cells

Macrophages express the signal-regulatory protein α, which is composed of an extracellular immunoglobulin domain for ligand binding and a cytosolic domain that includes an ITIM for inhibitory signaling [57,58,59,60,61]. CD47 was later identified as the ligand for SIRPα, and the CD47- signal-regulatory protein α pathway serves as a “don’t eat me” signal, resulting in the prohibition of signal-regulatory protein α-expressing macrophage phagocytizing CD47-expressing cells [58,59,60,61]. The CD47–signal-regulatory protein α axis prohibits the uptake of cancer cells by APCs, which eventually leads to failure in the establishment of anti-cancer adaptive immunity. Thus, we can regard CD47 as an important immune checkpoint molecule. Similar to the PD-1/PD-L1 and CTLA4, CD47 antagonists were generated. Recently, it was reported that an anti-CD47 mAb capable of Fc-FcγR interaction between the mAb and macrophages enhances anti-cancer activity in vivo [62]. Osorio et al. generated an anti-CD47 mAb with Fc variants having different affinities for FcγR, and found that treatment with the high-affinity variant highly protected mice bearing CD47^+^ cancer cells compared with treatment with the low-affinity variant. Fewer CD4^+^Foxp3^+^ Tregs accumulated in treated mice. This is presumably attributable to the high levels of CD47 expression in CD4^+^Foxp3^+^ Tregs. Thus, the CD47 antagonist was selected as the second ICI. CD47 antagonists inhibit cancer growth by restoring and enhancing APC-mediated innate and adaptive anti-cancer immunity.

## 4. CIRT Activation of Anti-Cancer Immunity

Since heavy-ion irradiation does not impair healthy tissues beneath cancer due to its unique property named spread-out Bragg peak, as described above, it is plausible that the immune system remains healthy in response to the cancer fragments or particles generated by CIRT. Indeed, ICRT-irradiated cancer cells harbor robust cytotoxic lymphocytes such as CD8^+^ T cells and NK cells, which are rarely observed in non-irradiated cancer cells [63,64]. However, the underlying mechanisms are not fully understood. Recently, it was verified that, in response to cytoplasmic DNAs liberated from CIRT-damaged cellular nuclei and mitochondria, cytoplasmic DNA sensor signaling triggers inflammatory innate and cancer-specific adaptive immune responses by inducing type I IFN secretion. In this section, we describe this pathway and other pathways involved in the activation of innate immunity. The activation of these pathways is linked to the activation of anti-cancer adaptive immunity. Many constructive review articles provide detailed narrations of breakthrough findings [64,65,66].

### 4.1. Cancer Cell Responses to Carbon Ion Irradiation

#### 4.1.1. Type I IFN (IFN-I) by Cyclic GMP–AMP Synthase (cGAS)-Stimulator of Interferon Genes (STING) Signaling

As mentioned above, CIRT-induced DNA damage in cancer cells results in the release of clustered DNA breaks from the compartmentalized nuclei and mitochondria into the cytosol. These cytoplasmic dsDNAs are recognized by DNA sensors, namely, cGAS, which eventually activates the STING pathway [63,67,68,69,70]. Secondary messengers of the cGAS-STING-mediated pathways are highly conserved in the Kingdom of Life [71]. The cGAS-STING signaling evokes IFN-I and various proinflammatory cytokines/chemokines, mainly by activating interferon-regulatory factor (IRF3) and nuclear factor κB, respectively. 

##### Cytotoxic and Cytoprotective Actions of IFN-I 

Many early studies have revealed IFN-I as a potent anti-cancer molecule in the context of its ability to induce cell growth arrest or the death of cancer cells. Based on these findings, IFN-I, both natural and recombinant, was successfully administered to patients with limited malignancy, such as hairy cell leukemia, melanoma, and renal cancer [72,73]. However, IFN-I is now replaced by new anti-cancer regimens. This may be due to the complex functions of IFN-I, as described below.

IFN-I continues to attract the attention of biologists. Many recent studies have revealed that endogenous IFN-I is necessary for the anti-cancer action of novel strategies [74,75,76]. Herein, we describe this issue in terms of the anti-cancer efficacy of CIRT. Second, IFN-I has unique signaling properties and biologically different downstream outcomes [77]. After the stimulation of cancer cells with IFN-I, the transcription factors signal transducer and activator of transcription (STAT)1 and STAT2 are phosphorylated, followed by heterodimer formation with phosphorylated STAT1 and STAT2, which, in combination with IRF9, translocates to the nucleus to activate its corresponding gene expression. This signaling event ends shortly, presumably because of the prompt elimination of harmful phosphorylated STAT1 and STAT2. However, unphosphorylated STAT1 and STAT2 continue to form a complex with IRF9, which activates the same transduction, poorly but persistently. Cancer cells with IFN-I-related gene signatures are resistant to radiotherapy-induced DNA damage [78,79]. Cancer cells harboring the unphosphorylated STAT1/unphosphorylated STAT2/IRF9 complex are more resistant to DNA damage-inducing chemotherapy than those lacking the complex. Furthermore, human cancer cell lines expressing STAT2, a constituent of the complex, are more resistant to radiotherapy than those that do not express *STAT2*. This is also true of patients with cancer. Patients with cancers that express rich *STAT2* have a more impaired prognosis than those expressing poor *STAT2* [80]. Recently, the mechanism underlying this phenomenon was partially unveiled [81]. STAT2 binds to STING, which localizes near the ER and prohibits the translocation of STING into the cytosol upon DNA damage-inducing agents. Therefore, one may assume that *STAT2* is a clinical marker of CIRT-resistance.

##### IFN-I Triggers Innate Immune Responses

IFN-I can activate dendritic cells (DC) and professional APCs to express the co-stimulators CD80 and CD86, which are necessary for T cell activation, and increase their cross-presentation ability. Extracellular or phagocytosed antigens are associated with MHC class II molecules, whereas cytoplasmic antigens are associated with MHC class I molecules. This difference is based on the antigen-processing sites in APCs. Furthermore, CD4+ T cells recognize antigens in association with MHC class II, whereas CD8^+^ T cells are activated only when antigens are presented in MHC class I. Therefore, extracellular or phagocytosed antigens usually cannot activate CD8^+^ T-cells. DCs are professional APCs because of their ability to cross-present extracellular antigens in association with MHC classes I and II. IFN-I activates the cross-presenting ability of DCs [82,83]. Thus, cancer cell fragments generated by carbon ion irradiation induce cancer-specific CD8^+^ T cells through the cross-presentation of CD80 and CD86 on DCs. IFN-I, produced by the cGAS-STING pathway, also enhances NK cell cytotoxicity. 

#### 4.1.2. High-Mobility Group Box 1 (HMGB1)

Upon CIRT, cancer cells are damaged and release HMGB1, which is classified as an alarmin that is a cytosolic molecule in intact cells that is released upon cellular damage. HMGB1 is regarded as a danger-associated molecular pattern and is recognized by Toll-like receptor-4 [84,85]. Previous studies have shown that CIRT-damaged cancer cells participate in anti-cancer immunity by activating HMGB1/Toll-like receptor 4-mediated inflammatory responses.

#### 4.1.3. Extracellular Expression of Heat Shock Proteins 70 

Heat shock proteins are upregulated by various types of cell stressors, such as heat- and radiotherapy-induced ROS. Under normal conditions, cancer cells produce high levels of heat shock proteins 70. Irradiation further upregulates heat shock protein 70 expression via ROS generation [86,87]. Irradiated cancer cells also express heat shock protein 70 extracellularly. Extracellular heat shock proteins 70 can activate NK cells to proliferate and migrate into damaged cancer cell clusters [87].

#### 4.1.4. PD-L1 Induction

After CIRT, cancer cells undergo clustered DNA breaks that activate the DNA repair response. The DNA repair process activates PD-L1 through IRF1-mediated STAT1 and STAT3 phosphorylation [88]. PD-L1 expression levels determine the responsiveness to ICIs [89]. Thus, CIRT-induced, dsDNA repair-mediated PD-L1 induction renders cancer cells susceptible to PD-1/PD-L1 treatment.

### 4.2. CIRT Activation of Innate Immunity

Dying cancer cells induced by CIRT are the major source of DNA. After the internalization or phagocytosis of DNA or dying cells, the cGAS-STING pathway is activated in the cytosol of phagocytes [63,67,68,69,70]. In line with the cancer cells, phagocytes in the vicinity of the cancer cells likely also secrete IFN-I. 

IFN-I derived from the cGAS-STING axis, either in cancer cells or DC, is capable of activating NK cells, DCs, and macrophages [74], converging on local inflammatory responses in cancer. Mice deficient in STING or STAT1 downstream factors activated by IFN-I show impairments in these immunological responses [90], verifying the importance of STING and IFN-I signaling.

In response to HMGB1 released by dying cancer cells, macrophages produce proinflammatory cytokines, such as IL-6 and TNF, via the Toll-like receptor 4-mediated pathway. Inflammation-prone conditions may trigger an inflammation-shifted TME, which may promote anti-cancer progress [91,92,93].

Extracellular heat shock proteins 70 expressed on irradiated cancer cells directly activate and recruit NK cells, which may participate in cancer cell killing [87]. 

Carbon ion irradiation reduces MDSC numbers in melanoma-bearing mice, although the mechanism underlying this reduction is unknown [94]. 

### 4.3. CIRT Induction of Cancer Cell-Specific Adaptive Immunity

Carbon-ion irradiation insults cancer cells, which then activate anti-cancer adaptive immunity (Figure 2). As mentioned above, carbon ion-irradiated cancer cells undergo various biological processes. A high linear energy transfer of CIRT damages the cancer DNA into dsDNA breaks, causing more complicated DNA damage. DNA repair activation is likely insufficient to restore damaged DNA, resulting in the programmed cell death of cancer cells. Cancer cell fragments and particles are phagocytosed by phagocytes including immature DCs. Simultaneously, cancer cell-derived dsDNA activates the DC cGAS-STING axis to release IFN and other cytokines by activating the IRF1- and NF-κB-mediated signaling pathways, respectively [68]. In addition, immature DCs take up cancer cell-derived dsDNA, leading to the activation of their own cGAS-STING axis within immature DCs. In response to these cGAS-STING pathway-mediated cytokines, cancer cell fragments/particle-internalized immature DCs differentiate into mature DCs that express CD80 and CD86, as well as robust MHC class II, and exert cross-presentation activity. The internalized cancer cell fragments/particles are processed into peptides. Mature DCs present and complement these cancer-derived antigen peptides in association with MHC class II and class I molecules, respectively. Furthermore, mature DCs express chemokine (C-C motif) receptor 7 [95], which allows them to migrate into the T-cell zone of draining lymph nodes rich in chemokine (C-C motif) ligand 19 and CCL20, which are ligands of CCR7 [96]. Thus, in the T-cell zone, cancer-associated antigen-specific naïve CD8^+^ and CD4^+^ T cells are fully activated and become cytotoxic and effector cells, respectively. These activated lymphocytes egress the LN and enter the cancer, where they exert their anti-cancer functions.

### 4.4. Radiotherapy Activation of Anti-Immune Response

As described in Section 4.2, CIRT indirectly and locally induces cancer cell death through the activation of innate immunity. This cancer cell death is designated as immunogenic cell death (ICD) [2,4,9,97]. ICD induced by the innate immune response contributes to the activation of systemic anti-cancer adaptive immunity, as described in Section 4.3. This systemic anti-cancer immunity has potential to trigger abscopal response, which is defined as the regression of cancer mass away from the mass having received radiotherapy [98,99]. It is well documented that systemic anti-cancer immune responses, including the induction of ICD and abscopal response, develop upon photon and proton radiotherapies as well as CIRT [2,4,9,97,98]. Thus, anti-cancer immune responses commonly develop upon radiotherapy, which eventually results in cancer growth impairment and happens to lead to cancer eradication. 

There is only one article that reported the differences in the activation of innate immune responses between irradiation with photons, protons, and carbon ions [90]. Du et al. irradiated esophageal cancer cells with X-rays, protons, and carbon ions in vitro and conducted RNA-sequencing of the cancer cells. They found that irradiation-induced innate immune responses were comparable between the three types of radiotherapy [90]. It remains unclear, however, whether photon, proton, and carbon irradiations induce different intensities of adaptive anti-cancer immune responses against the same type of cancer cells implanted on mice. 

Based on the immunological processes described in Section 3 and Section 4, one may assume that CIRT-induced anti-cancer adaptive immunity is enhanced by diminishing the cancer-promoting action of the TME. 

## 5. Radiotherapy Induction of Abscopal Effects in Combination with Immunotherapy

### 5.1. Radiotherapy Together with ICIs Induces Abscopal Effects and Establishes Immunological Memory against Cancer 

#### 5.1.1. CIRT

The possible scenario described at the end of the previous section was verified, at least in part, in cancer-bearing mice. Carbon ion irradiation in combination with CTLA4 blockade has been reported to rescue mice from tumors. Both legs of the mice were inoculated with murine osteosarcoma cells. After mass development, the mice were systemically administered anti-CTLA4 mAb or anti-PD-L1 mAb, followed by CIRT on one side of the tumor. Treatment with CIRT together with CTLA4 blockade, but not with PD-L1 blockade, protected the mice from tumor growth on both sides, distal tumor metastasis, and mortality [6]. This suggests that CIRT in tumors may initiate a systemic anti-osteosarcoma adaptive immune response, which eventually alleviates tumor growth on both sides. The antitumor efficacy of this combination was higher than that of CIRT or CLTA4 blockade alone. Recently, consistent findings were reported in mice with breast cancer [7]. Hartmann et al. subcutaneously inoculated syngeneic mouse breast cancer cells into the right hind leg and left flank. After the cancer mass was established, CIRT was performed only on the right leg tumor, followed by a systemic injection of ICIs against CTLA4 or PD-LA. They found that a combination treatment with carbon ion irradiation and anti-CTLA4 mAb, but not anti-PD-L1 mAb, protected against cancer growth on both sides. This was also observed for survival. Mice treated with CIRT alone or with CIRT plus anti-PD-L1 blockade exhibited survival data comparable to those of breast cancer-bearing mice without any treatment, and the mice in all three groups died within 40 days. In contrast, mice receiving both CIRT and CTLA4 blockade show a much higher survival rate, with more than half of the mice surviving for 100 days and more. Furthermore, surviving mice completely rejected breast cancer after reinoculation [7]. Huang et al. recently reached the same conclusion using melanoma-bearing mice [100]. This indicates that combination therapy with carbon ion irradiation and CTLA4 blockade established a memory response against breast cancer. The immunological memory established by ICRT with ICIs might protect against the future metastasis and recurrence of cancer.

The ICIs currently available produce severe adverse effects, such as the induction of autoimmunity. To exclude this harmful response, we need to generate cancer Treg-specific ICIs. As described in Section 3.1.1, windows are opening for medicines selectively targeting cancer Tregs. This might link to the generation of cancer-specific ICIs in the near future.

#### 5.1.2. Photon Radiotherapy and Proton Radiotherapy

Photon irradiation in combination with immunotherapy also has the potential to induce abscopal effects on tumor-bearing mice. Demaria et al. reported that treatment with photon irradiation and ICI protected against primary tumor growth and metastases in a murine model of breast cancer [101]. They found anti-cancer effects of γ-irradiation in combination with anti-CTLA4 mAb on breast cancer, with protection against metastases. Zahidunnabi Dewan et al. reported immunological memory against mammary carcinoma in murine mammary carcinoma-bearing mice after treatment with γ-irradiation together with anti-CTLA4 mAb [102]. Consistently, Twyman-Saint Victor et al. clearly demonstrated that photon radiotherapy in combination with CTLA4 and PD-L1 blockades activates abscopal effects and establishes immunological memory against melanoma in a mouse model of melanoma. Notably, they showed that CTLA4 and PD-L1 blockades activate non-redundant mechanisms in the melanoma of mice receiving photon radiotherapy: CTLA4 blockade decreases and increases the proportion of Treg and CD8^+^ T cells in tumor-infiltrating lymphocytes, respectively, but simultaneously induces exhaustion in CD8^+^ tumor-infiltrating lymphocytes via inducing PD-L1 expression on cancer, while PD-L1 blockade reinvigorates these exhausted CD8^+^ T cells. Plausibly, the dual blockade of CTLA4 and PD-L1 together with irradiation converges on the cancer regression with abscopal responses [103]. Proton radiotherapy together with PD-L1 blockade was also reported to induce abscopal responses in a mouse model of hepatocellular carcinoma [104]. Thus, radiotherapy, at least photon, proton, and carbon ion radiotherapy, in collaboration with ICIs has potential to achieve their anti-cancer activities and perhaps to establish immunological memory against cancer.

### 5.2. Novel Modalities against MDSC and TAM

As described in Section 3, the TME consists of pro-cancerous MDSCs and TAMs as well as Tregs. Novel and potent therapies have been developed recently: anti-MDSC therapy via IL-4 signaling blockade [52] and anti-TAM therapy via second-generation M1-polarized CAR macrophages [56]. Therefore, radiotherapy combined with IL-4 signaling inhibitors and/or second-generation M1-polarized CAR or in combination with medicine targeting cancer Tregs [38] might become a next-generation regimen for radioimmunotherapy.

## 6. Clinical Trials of CIRT in Combination with ICIs

Immunoradiotherapy, a combination of photon radiotherapy and ICIs, was performed several years ago with substantial success. Patients with stage III non-small-cell lung cancer were administered mAbs against PD-L1 or a placebo after chemoradiotherapy. Patients treated with an anti-PD-L1 antibody showed longer progression-free survival than those treated with a placebo [105]. Furthermore, patients who received anti-PD-L1 therapy exhibited longer overall survival than the control patients [106]. The anti-cancer efficacy of ICIs has been summarized in recent reviews [107,108]. Based on these reports, we assume that CIRT combined with ICI treatment may be beneficial for cancer therapy. However, only one trial (NCT05229614) was included [109]. We hope that this clinical trial will be successful.

## 7. Conclusions

CIRT exerts its anti-cancer action by activating anti-cancer innate and adaptive immunities as well as its direct cytocidal action on cancer cells. To achieve its anti-cancer function, radiotherapy including CIRT requires collaboration with modalities critically and selectively targeting Tregs in the TME. The combination with modalities targeting Tregs, MDSCs, and TAMs will become a future regimen for cancer eradication. 

## 8. Future Prospects 

Cancer immunotherapies are classified into two groups based on the features of the anti-cancer-specific T cells involved. One type are T cells that express engineered exogenous T-cell receptors that recognize cancer-associated antigens in association with self-MHC. Since neoantigen-targeting technology has been promptly developed, we currently have diverse clinical trials targeting human solid cancers, such as CAR-T cells [110,111,112]. The second module activates endogenous anti-cancer immunity, such as ICIs, along with radiotherapy, including CIRT, as described in this review. DC transfer therapy activates endogenous anti-cancer immunity [113]. Recently, however, the mitochondrial antiviral signaling protein (MAVS), an important signaling molecule in innate immunity, was shown to downregulate the anti-cancer immunity of DCs [114]. Members of the retinoic acid-inducible gene I-like receptor family and melanoma differentiation-associated gene 5 commonly utilize MAVS to activate their signaling for viral eradication [115]. In contrast, non-canonical MAVS signaling alleviates DC-mediated anti-cancer immunity. From this report, we assume that the transfer of MAVS-deficient DC instead of wild-type DC might induce robust anti-cancer immunity in CIRT-receiving patients with cancer.

Although the activation of autophagy is regarded as a trigger of cell death [116], certain types of cancers, such as pancreatic cancer and intrahepatic cholangiocarcinoma [117,118,119], require autophagy to survive under hypoxic and nutrient-poor conditions. CIRT has been reported to activate autophagy. Therefore, it is plausible that autophagy-dependent cancers evade the cytotoxic effects of CIRT [2,120]. To achieve the full anti-cancer efficacy of CIRT-mediated immunotherapy, autophagy-dependent cancers might require additional supplementation with autophagy antagonists. The combination treatment of pancreatic cancer cells with CIRT and the autophagy inhibitor hydroxychloroquine promotes pancreatic cancer cells both in vitro and in vivo [117]. Therefore, autophagy signaling is the third target of CIRT-mediated immunotherapy for autophagy-dependent cancers.

## Figures and Tables

**Figure 1 ijms-25-02830-f001:**
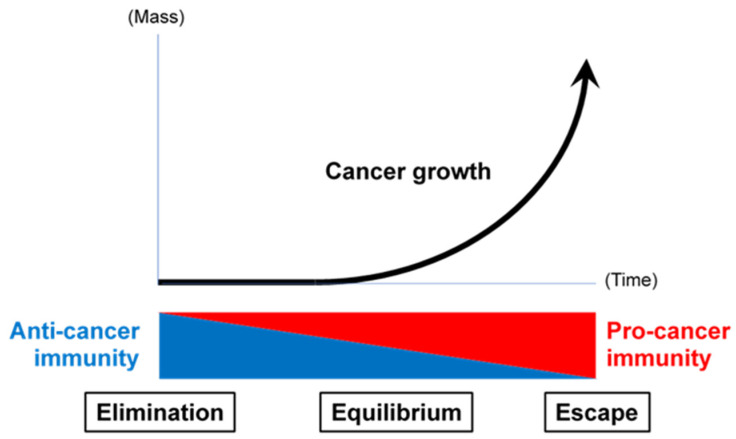
Cancer immunosurveillance. Anti-cancer immunity and pro-cancer immunity regulate cancer development, reciprocally.

**Figure 2 ijms-25-02830-f002:**
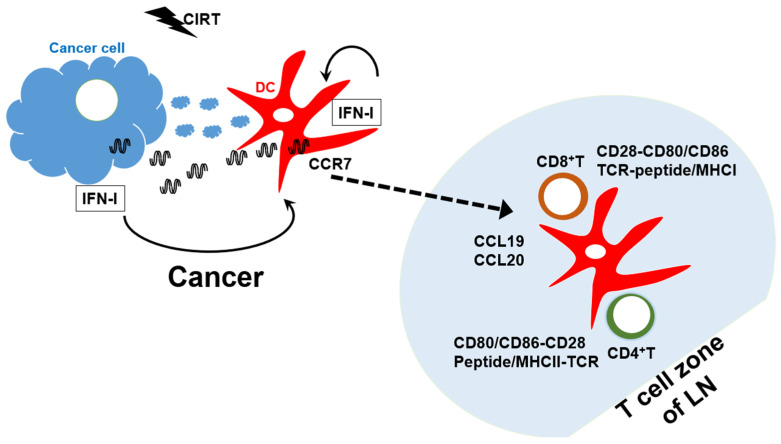
CIRT-mediated cancer immunotherapy. Upon carbon ion irradiation, cancer cells undergo double-strand (ds) DNA breaks in nuclei and mitochondria, resulting in the cytoplasmic translocation of dsDNA. Cytoplasmic dsDNA activates the cGAS-STING axis, eventually leading to the release of type I IFN (IFN-I). Immature dendritic cells engulf and phagocytize dead cancer cells and their dsDNA, activating the cGAS-STING pathway in immature DCs. These events converge on the DC maturation, in terms of the capacity to present cancer antigens to the corresponding T cells. Mature DCs capture and process cancer antigen (Ag) and migrate into the T cell zone of draining lymph node (LN) via the interaction of their CCR7 with its ligands CCL19 and CCL20 in the T-cell zone. In the T-cell zone, DCs activate Ag-specific CD4^+^ T cells via the interaction of CD80/CD86 with CD28, as well as the interaction of Ag peptide on MHC class II with Ag-specific T-cell receptor. Mature DCs also activate CD8^+^ T cells through the cross-presentation of Ag in association with MHC class I. The activated Ag-specific CD8^+^ cells and CD4^+^ T cells egress LN and enter the cancer, where these two types of T cells exert their cancer-killing activity and effector function, respectively.

## Data Availability

No new data were created or analyzed in this study. Data sharing is not applicable to this article.

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
