# Peer review of "Carbon Ion Irradiation Activates Anti-Cancer Immunity"

_ijms, 2024, doi:10.3390/ijms25052830_

Round 1

Reviewer 1 Report

Comments and Suggestions for Authors

The topic chosen by the authors for this review is arguably of interest since it combines carbon-ion therapy, which is the most effective form of hadrontherapy used at the moment against radioresistant cancers, and the recently (re)discovered potential by ionising radiation to elicit an immune response with tumoricidal effects,

However, there are some criticalities in the layout of this work that require accurate revision before being considered for publication.

First of all, the technical language used by the authors to describe the radiobiological action of heavy charged particles such as carbon ions is at times too simplistic: This is the case in the Introduction  in the opening sentence "Carbon ion beams have the unique property of selectively providing high-dose radiation  in cancer without causing adverse effects on healthy tissues around cancer. This property evokes  abundant double-strand or complicated DNA breaks, which allow the cytosolic translocation...". Protons do have the same ballistic precision, what makes carbon ions more biologically effective is their higher linear energy transfer (LET), that is the capacity of causing not more abundant breaks per a give dose of photons or protons, but of a greater complexity. And whether the damage induced in healthy tissue is not proven really true because of the higher RBE of carbon ions whose LET is higher than that of photons and protons also in the beam entrance, plus the problem of fragmentation beyond the Spread-Out Bragg peak (which the authors simplistically refer only to as Bragg peak).  This sentence is exemplary because in the review one of putative reasons why carbon ion irradiation seems more effective than other radiotherapy modalities at activating immune response is the known cGAS-STING mediated therapy. This pathway seems to require release in the cytoplasm of DNA breaks but the authors state that carbon ion-induced breaks translocate form the nucleus and/or mithocondria and hence they activate the pathway. However, a review should be critical and discuss findings such as these: we know that the main reason of carbon ion superiority at killing (cancer9 cells is their ability to cause clustered damage which  impacts the cell repair system. Yes, this sometimes triggers apoptosis and release in the cytoplasm of DNA fragments but the picture is more compilcated than this, or at least there should be other mechanisms underlying a putative superiority of carbon ions also at triggering immune response. I shall come back to the lack of some features a review should contain.

First, I would like to convey to the authors also my disagreement on the Section 3 on Cancer Immunosurveillance, which is unnecessarily long in my view, with an excess of in-depth description of mechanisms which should be cited only is really instrumental for the focus of the review. And we come back to the core issue: the part referring to carbon ions per se is quite succint, a list of results without a section of Discussion or Conclusion and without citing mechanisms and phenomena which are also been known to be associated with radiation-induced upregulation of immune response: immunogenic cell death (ICD), abscopal response (only briefly hinted at when talking on in vivo studies). Hnece, one would expect a critical discussion of the findings reported on carbon ion irradiation compared, for instance, with those with protons, speculating on the differences (if any) and articulating more in depth on the vast and exciting prospective of combining radiotherapy/hadrontherapy with immunotherapy, which semms here to be related only to one cell cyle inhibitor. Moreover, even assuming that the authors would prefer a focus exclusively on carbon ion without talking of other less efficient modalities of radiotherapy on the immune system, one would expect some insights on what the best doses are to elicit the maximum effect, according to what temporal regimes they should be administered and so on.

Comments on the Quality of English Language

Please revise especially the English language when referring to specific radiobiological mechanisms because it lack of accuracy.

Reviewer 2 Report

Comments and Suggestions for Authors

1. The description of the direct background that led to the writing of this review paper is very poor. Line 27 to 42 are general information about particle beam therapy and are not the direct background of this review article. The contents of lines 43 to 45 can be seen as the direct background of this paper, but they are too sparse. The description of the direct background of this review article needs to be expanded.

2. Too many abbreviations were used throughout the paper. The use of abbreviations should be minimized, except for widely known and frequently used abbreviations.

3. Authors must add references to the contents of lines 358 to 363.

4. 6.1 Clinical studies of CIRT (Line 434-464)

This content does not fit the purpose of this review article. The main purpose of this paper is to explain how CIRT reacts with host immunity. Lines 434-464 summarize recent clinical trials of CIRT and do not meet the purpose of this paper. It would be better to delete it. 

Round 2

Reviewer 1 Report

Comments and Suggestions for Authors

I thank the authors for providing a well-revised version of their work, whose publication in IJMS I feel to recommend.